# System Theoretical Study on the Effect of Variable Nonmetallic Doping on Improving Catalytic Activity of 2D-Ti_3_C_2_O_2_ for Hydrogen Evolution Reaction

**DOI:** 10.3390/nano11102497

**Published:** 2021-09-25

**Authors:** Ye Su, Minhui Song, Xiaoxu Wang, Jihang Jiang, Xiaolong Si, Tianhang Zhao, Ping Qian

**Affiliations:** Beijing Advanced Innovation Center for Materials Genome Engineering, School of Mathematics and Physics, University of Science and Technology Beijing, Beijing 100083, China; suyechina@163.com (Y.S.); mhuisong@163.com (M.S.); xuying13520506589@163.com (J.J.); sixiaolong2018@outlook.com (X.S.); ztx13520805687@163.com (T.Z.)

**Keywords:** nonmetallic doping, 2D-Ti_3_C_2_O_2_, DFT, HER

## Abstract

2D MXenes have been found to be one of the most promising catalysts for hydrogen evolution reaction (HER) due to their excellent electronic conductivity, hydrophilic nature, porosity and stability. Nonmetallic (NM) element doping is an effective approach to enhance the HER catalytic performance. By using the density functional theory (DFT) method, we researched the effect of nonmetallic doping (different element types, variable doping concentrations) and optimal hydrogen absorption concentration on the surface of NM-Ti_3_C_2_O_2_ for HER catalytic activity and stability. The calculation results show that doping nonmetallic elements can improve their HER catalytic properties; the P element dopants catalyst especially exhibits remarkable HER performance (∆GH = 0.008 eV when the P element doping concentration is 100% and the hydrogen absorption is 75%). The origin mechanism of the regulation of doping on stability and catalytic activity was analyzed by electronic structures. The results of this work proved that by controlling the doping elements and their concentrations we can tune the catalytic activity, which will accelerate the further research of HER catalysts.

## 1. Introduction

Excessive exploitation and consumption of fossil fuels and the greenhouse effect caused by excessive emission of carbon dioxide have brought ineluctable environment problems [1]. Given this challenge, clean and renewable energy sources are urgently needed. Hydrogen energy is considered as one of the most promising substitutions for fossil fuels due to its eco-friendliness, abundant, high energy density and utility value [2]. Various methods have been developed to generate hydrogen [3,4,5], among which electrocatalytic water splitting is considered to be the cleanest and one of the most effective methods for the massive production of hydrogen [6,7,8]. Due to the overpotential of HER, it limits its practical application, it requires ideal catalysts to accelerate the reaction process [9].

Currently, platinum-based materials are the most effective catalysts and large-scale applied in technology [10]. However, its natural scarcity, high price and strong absorption of CO on the active surface site of Pt restricts its practical utilization. Therefore, it is desirable to research and develop high-performance electrocatalysts which are cost-effective and not poisonous. In the past few decades, many alternatives have been sought, including non-precious metal alloys [11], metal chalcogenides [12,13,14], metal carbides [15,16,17], metal nitrides [18], metal phosphides [19] and so on [20,21]. Among them, two-dimensional transition metal carbides/nitrides (MXenes) attracted intense attention for the application of HER catalysts.

MXenes, a family of 2D layered materials, which have the general formula M_n+1_X_n_T_x_, where M stands for early transition metals, X stands for C or N, T_x_ are the surface functional groups O, OH, or F, and n = 1, 2 or 3 [22]. MXenes are usually synthesized by chemical etching A layers in MAX phase to expose the reactive metal atoms, which would immediately become functionalized with surface terminations (e.g., O, OH, F) [23]. MXenes are finding their way into a myriad of applications including electrocatalysis [24], hydrogen storage [25,26], lithium-ion batteries [27,28] and supercapacitors [29]. Owing to the existence of surface terminations, MXenes have high surface area, excellent electronic conductivity, hydrophilic nature, porosity and stability. Therefore, in the process of HER reaction, MXenes can generate new active sites and expand the reaction area, thus promoting the interaction between hydrogen and MXenes [30]. Although many previous theories and experiments confirmed that MXenes are promising catalysts for HER, they are still far from sufficient for practical applications. In order to realize possible practical usage, some methods were required to further enhance the HER catalytic activity of MXenes. 

The performance of MXenes can be fine-tuned by surface absorption and doping. According to the report, by substitution of Mo with Co in the Mo_2_CT_x_ lattice, MXenes displayed an impressive durability and activity, and activity and stability for HER can be enhanced by N atom doped Ti_3_CT_x_ [31]. Significantly, doping concentration and hydrogen absorption concentration also have large effects on the catalytic activity. It has been reported that by high N atom doping, MoS_2_ catalyst can improve the electrical conductivity, thus facilitating the HER activity [32]. The latest experimental research found that doping low concentration Co can enhanced MoSe_2_ catalytic activity toward HER [33]. Recently, it was found that the optimum evolution of H_2_ occurred on the B-terminated {001} surface of MoB_2_ when hydrogen coverage is between 75% and 100% [34].

It has been confirmed by experiments that doping improves HER performance. Due to the variety of nonmetallic elements and the diversity of MXene materials, it is difficult to screen out materials with excellent performance and systematically research only by experimental means. Theoretical calculation as a complement to the experimental, can research the intrinsic origin of the HER activity, provide a wealthy data base, so as to systematically and comprehensively screen MXene materials and predict potential HER candidates. 

In this work, we select 2D-Ti_3_C_2_O_2_ as the object investigating the stability and catalytic activity of six kinds of nonmetal element doping by DFT theoretical calculation. Firstly, we studied the structure and stability of the Ti_3_C_2_O_2_ with different nonmetals types and doping concentrations. We then investigated the catalyst activity of doping structure with different absorption concentrations and configurations. Further, the intrinsic relationship between the catalytic activity and effect of doping is explored from the perspective of the electronic structure analysis. Finally, we verify whether the candidate catalyst can be synthesized by experiment.

## 2. Materials and Methods

The first-principles simulation of density functional theory (DFT) is carried out using the Vienna ab initio simulation package (VASP) [35,36], the projection enhanced wave (PAW) method [37] and the generalized gradient approximation (GGA) are used in the Perdew–Burke–Ernzerhof (PBE) form [38,39]. The cut-off kinetic energies for the plane waves are set to 600 eV for all the calculations. In the process of structural relaxation, the energy and force convergence condition are 1.0 × 10^−5^ eV and 0.01 eV/Å. According to the empirical correction in Grimme scheme, DFT+D3 [40] considers van der Waals interaction, and at least 20 Å vacuum spaces are used to avoid the artificial interaction between periodic elements. The Brillouin region of 2 × 2 × 1 supercell was sampled by using a 9 × 9 × 1 uniform k-point grid. It is known that the O-terminal is produced by hydrofluoric acid etching, and the catalytic activity was proved to be better than other passivating groups, such as F [41]. In this work, 2D Ti_3_C_2_O_2_ are limited to research, whose surface functional groups are all composed of O atoms.

The substitution energy of different nonmetallic doping elements can be defined by Equation (1):(1)$${\Delta E}_{\mathrm{sub}\left(i\right)}=\left(E_{O_{1-n_{i}}X_{n_{i}}/{\mathrm{Ti}}_{3}C_{2}O_{2}}+{\mathrm{iE}}_{O}\right)-\left(E_{{\mathrm{Ti}}_{3}C_{2}O_{2}}+{\mathrm{iE}}_{X}\right)$$
where X is the nonmetallic element (B, C, F, N, P and S).${\;E}_{O_{1-n_{i}}X_{n_{i}}/{\mathrm{Ti}}_{3}C_{2}O_{2}}$ is the total energy of the Ti_3_C_2_O_2_ surface with the nonmetallic element X replacing the O atom; $E_{{\mathrm{Ti}}_{3}C_{2}O_{2}}$ is the total energy of the Ti_3_C_2_O_2_ surface. $E_{O}$ and${\;E}_{X}$ are the energies of one O atom and one nonmetallic X atom, respectively, and i is the number of nonmetallic X atoms substituted by doping in the model. Additionally, n_i_ is the doping substitution concentration of nonmetallic elements.

Adsorption energy of H atoms on substituted O surface by doping nonmetallic elements by Equation (2):(2)$${\Delta E}_{H}=E_{O_{1-n_{i}}X_{n_{i}}/{\mathrm{Ti}}_{3}C_{2}O_{2}+n_{j}H}-\left(E_{O_{1-n_{i}}X_{n_{i}}/{\mathrm{Ti}}_{3}C_{2}O_{2}}+\frac{j}{2}E_{H_{2}}\right)$$
where $E_{O_{1-n_{i}}X_{n_{i}}/{\mathrm{Ti}}_{3}C_{2}O_{2}+n_{j}H}$ is the total energy of H atom adsorbed on the surface of Ti_3_C_2_O_2_ after doping and substitution of nonmetallic elements. $E_{H_{2}}$ is the energy of one H molecule and j is the number of H atom adsorbed in the model. Additionally, n_j_ is the adsorption concentration of H atoms.

Gibbs free energy of hydrogen adsorption (∆G_H_) is defined by Equation (3) as follows:(3)$${\Delta G}_{H}={\Delta E}_{H}+{\Delta E}_{\mathrm{zep}}-{T\Delta S}_{H}$$
where ${\Delta E}_{H}$ is the adsorption energy of a H atom adsorbed on the surface of O element doped by nonmetallic elements and ${\Delta E}_{\mathrm{ZEP}}$ is the zero potential energy correction.

The $\Delta S_{H}$ can be approximated by Equation (4):(4)$$\Delta S_{H}≅-\frac{1}{2}S_{H_{2}}^{0}$$
where $S_{H_{2}}^{0}$ is the entropy of H_2_ gas under the standard condition due to the fact that the vibrational entropy in the adsorbed state is small according to the previous studies [42] and $S_{H_{2}}^{0}$ is the entropy of H_2_ gas under the standard condition. Here, the values of ∆E_ZPE_ and T∆S_H_ are referenced from reference [43]. Therefore, Equation (3) can be written as Equation (5).
(5)$$\Delta G_{H*}=\Delta E_{H}+0.3\;\mathrm{eV}$$

The optimal $\Delta G_{H}$ value for HER is close to 0 eV, which means that the smaller the value of$\;\left|\Delta G_{H}\right|$, the better the HER performance of the catalyst will be.

## 3. Results

### 3.1. Geometries and Thermal Stability

As one of the most common 2D MXenes, bare Ti_3_C_2_ formed a hexagonal lattice displaying P$\bar{3}$*m*1 group symmetry. It is made of alternating five atomic layers of Ti-C- Ti-C-Ti, in which there are two exposed Ti layers, and each C atom bonds with six Ti atoms. The most stable configuration of different MXenes varies with the functional groups. Because the exposed metal atoms are electron donors, they prefer being functionalized by electronegative functional groups [43]. Past research has shown that O is thermodynamically preferred and the most favorable absorption site of H [44,45]. In our previous work, we studied the stable O-terminated Mxene configurations, and the most favorable Ti_3_C_2_O_2_ configuration is O atoms at face-centered cubic hollow sites on both sides, as shown in Figure 1a [46]. Therefore, only Ti_3_C_2_O_2_ with this configuration is discussed below. Additionally, Figure 2 represents the workflow of our study of the effect on nonmetallic doping and ideal hydrogen absorption concentration on the surface of NM-Ti_3_C_2_O_2_ for HER catalytic activity and stability.

In order to study the effects of different nonmetals and different doping concentrations on the crystal structure, electronic structure and hydrogen adsorption, 2 × 2 supercell 2D Ti_3_C_2_O_2_ model was built in this work. As shown in Figure 1b, the dopant model is formed by replacing surface O atoms with the nonmetallic atoms. Since each O bond on the surface is connected to three Ti bonds on the subsurface, there are four types of equivalent doping positions, numbered 1 through 4. Herein, the nonmetallic elements we selected is B, C, F, N, P and S, and varied the substitution concentration from 0.25 to 1.00. At the same doping concentration, due to the existence of equivalent substitution sites, nonmetallic atoms doped at different positions formed different dopant models (Appendix A).

We then considered the stable adsorption of hydrogen atoms with different absorption concentrations on the surface of dopant models. H atoms prefer being absorbed at the top site of O termination as long as the functional groups contain O [44], which is represented by S_1_ in Figure 1c. Therefore, for Ti_3_C_2_O_2_, each H bond on the top site is connected to one bond on the surface, which resulted in them also having four equivalent absorption positions. When O atoms are replaced by doping atoms, an equivalent absorption site is at the top site of doping atoms, which is represented by S_2_ in Figure 1c. Hence, H atoms are possibly absorbed on the S_1_ or S_2_ sites for dopant models. In all configurations, we consider the range of H absorption concentration from 0.25 to 1.00. At the same absorption concentration, the number of doping atoms and O atoms absorbed by H atoms may be different, which formed different absorption models (Appendix A).

Stability is one of the important parameters to evaluate the catalytic performance, more stable catalyst and better catalytic performance. The substitution energy(ΔE_sub(i)_) can indicate the stability of the surface. If the value of ΔE_sub(i)_ is negative, the structure is stable. On the contrary, if the value of ΔE_sub(i)_ is positive, the structure is unstable. In order to evaluate the stability of different doping models, we calculate the substitution energy of six nonmetallic elements (B, C, S, N, P and F) at different concentrations (0.25, 0.50, 0.75 and 1.00). It can be seen from Figure 3 that there are two different trends: (1) F have negative value; (2) B, C, S, N and P have all positive values. In addition to nonmetal F, the values of ΔE_sub(i)_ are all positive, and it is obvious that ΔE_sub(i)_ is closely related to the doping concentration (n_i_). With the increase of doping concentration, ΔE_sub(i)_ increased sequentially (0.25 < 0.50 < 0.75 < 1.00), which indicated that doping made the structure unstable, and the stability of the structure decreased with the increase of doping concentration. For element F, when the substitution concentration is 0.25, ΔE_sub(i)_ is negative, and the structural stability increases. As the doping concentration increases, ΔE_sub(i)_ increases, and the structural stability decreases. At all doping concentrations, the order of ΔE_sub(i)_ is ΔE_sub(F)_ < ΔE_sub(S)_ < ΔE_sub(N)_ < ΔE_sub(P)_ < ΔE_sub(C)_ < ΔE_sub(B)_. Among these elements, F is at the right of O on the periodic table, which has the least ΔE_sub(i)_; and S and O are in the same group, which has the next lowest ΔE_sub(i)_; and the value of ΔE_sub(i)_ increases in the same period with site deviation from O. This shows that the structure of F element doping is the most stable, and the structure of B element doping is the most unstable. At the same time, the trend we found is beneficial for us to select suitable doping nonmetallic elements when conducting experimental synthesis. Because of the poor configurational stability, in the following discussion, we will not consider the structure of B element doping.

### 3.2. Gibbs Free Energy of Hydrogen Adsorption

In order to further explore the nonmetal doping effects on the HER activity on the Ti_3_C_2_O_2_, the Gibbs free energy of hydrogen adsorption (∆G_H_) is calculated (Appendix A). Theoretically, the HER path can be described as containing the initial value of the initial state H + e–, the intermediate state of adsorption H* (* is the adsorption site), and the final state is 1/2 of the H_2_ product [9,47,48]. The ideal catalyst adsorption and desorption properties should not be too weak nor too strong. If the binding of hydrogen to the surface is too weak, the adsorption (Volmer) step will limit the overall reaction rate, while if the binding is too strong, the desorption (Heyrovsky/Tafel) step will limit the reaction rate. Therefore, a necessary, but not sufficient, condition for an active HER catalyst is ∆G_H_ ≈ 0 [42]. Therefore, the closer ∆G_H_ approaches 0, the better the catalytic performance. 

We first calculate ∆G_H_ of Ti_3_C_2_O_2_ surface with different H adsorption concentrations (1/4H, 2/4H, 3/4H and 4/4H represent 1, 2, 3 and 4 H atoms are adsorbed on the supercell surface, respectively). As shown in Figure 4a, the value of ∆G_H_ is −0.112 eV (1/2H), 0.024 eV (2/4H), 0.222 eV (3/4H) and 0.413 eV (4/4H). With the H absorption concentration increasing, the Gibbs free energy is increased. When 2 H atoms are absorbed on the surface, the value of |∆G_H_| is minimum, Ti_3_C_2_O_2_ has the best catalytic activity.

For the same dopant models, the number of H atoms absorbed on the surface ranged from 1 to 4 (representing the H absorption concentrations from 1/4 to 4/4). At the same H absorption concentration, H atoms are absorbed at different sites which formed different absorption models. We calculate ∆G_H_ of different absorption models (Appendix A). 

Additionally, at the same doping concentrations, we select the best value of ∆G_H_ to discuss. Figure 4b–f shows ∆G_H_ of the Ti_3_C_2_O_2_ doped with C, F, N, P and S at 0.25–1.00 concentrations.

In terms of the type of doping elements, the optimal order of |∆G_H_| is P (0.008 eV, at 1.00 concentration) > N (−0.028 eV, at 0.25 concentration) > S (0.060 eV, at 0.25 concentration) > C (−0.081 eV, at 0.25 concentration) > F (0.466 eV, at 0.50 concentration). The F doping model is the most table structure, but its catalytic effect is the worst. Additionally, the C doping model is the most unstable structure, but its catalytic effect is only better than F. For the remaining P, N and S elements, the catalytic effect is increased with the decrease of the stability of doping models. So too stable or too unstable models are not conducive to the improvement of catalytic activity. 

Except for F element, other element doping concentrations and H adsorption concentrations have significant effects on the catalytic activity. With the increase of doping concentration, the value of |∆G_H_| for C, N and S elements tends to rise, while P tends to decline. For C, N and P elements, the value of |∆G_H_| is generally decreased with the increase of H absorption concentration. Except for F element, when doping concentration is 0.25 and 1, the other elements have better catalytic effect when H atoms at high absorption concentration (3/4 and 4/4). Additionally, when doping concentrations are 0.25, C, N and S elements achieve the best catalytic effect.

### 3.3. The Descriptor and Origin of HER Catalytic Activity

In order to better understand the effect of nonmetallic element doping concentrations on ΔG_H_ and ΔE_sub_, we perform an exploratory research and analysis of the charge descriptor. Past research has shown that O atoms in different O-termination MXenes will gain different amounts of electrons from different metals due to their different electron-donating abilities. Therefore, the H-O bond strength and ΔG_H_ can be reflected by the Bader electron transfer of surface O atom gains (N_e_), a larger N_e_ will cause a weaker absorption of H on MXenes [49]. Figure 5 shows charge change after doping different concentrations of nonmetallic elements. It can be clearly seen from Figure 5 that the relationship between charge transfer and doping concentration is approximately linear. Two slopes can be observed: (1) slope doping F element is positive and (2) slope doping N, C, S and P is negative. Additionally, the slopes can indicate the direction of electron transfer between the subsurface Ti element and the doped nonmetallic elements. For F element, positive slope means electron transfer from Ti to F, oxidized Ti (electron donor) and reduced F (electron acceptor). On the contrary, for the other elements, a negative slope means the electrons move in the opposite direction. We find that for doping elements, the slope is positive when the electronegativity is greater than O, and negative otherwise. Besides, for elements whose electronegativity is less than O element, N and P, respectively, have the highest and lowest electronegativity, which means they have the highest and lowest electron acquisition capability. This matches with the charge change shown in Figure 5. However, there is no obvious trend between charge transfer and the change of ΔG_H_.

### 3.4. The Electronic Structure Insight of HER Catalytic Activity

The relationship between doping and catalytic activity cannot be appropriately described only by charge transfer. The possible reason is that doping not only distorted the local structure but also redistributed the overall electronic structure. Therefore, we take doping 0.25 concentration nonmetallic elements as an example, and discuss the variation of bond length (Appendix A) and the electron density difference (EDD). It can be clearly seen from Figure 6 that doping nonmetallic elements causes the change of bond lengths and electron distribution. With the increasing of Ti-NM bond lengths, the electrons gained by nonmetallic atoms decreasing and the interaction between Ti and nonmetallic atoms weakened, corresponding to the green weakening in Figure 6. For elements of the same period, such as N and P, for O and S the atomic number increases, the bond lengths become longer and the electrons gained decrease. The catalytic activity of Ti_3_C_2_O_2_ is regulated by the synergistic action of surface crystal structure and electronic structure.

In order to furtherly investigate the influence of the doping concentrations and H absorption concentration on the HER activity, the electronic density of states (DOS) and the difference charge density (DCD) of P doping Ti_3_C_2_O_2_ are systematically analyzed. Figure 7 shows the total density of states (TDOS) and projected density of states (PDOS) of Ti_3_C_2_O_2_ and Ti_3_C_2_O_2_ doping with different concentrations of P element with different H element absorption concentrations. Additionally, Figure 8 is the corresponding DCD. 

It is well known that when H atom absorbs on the surface of NM-Ti_3_C_2_O_2_, the interaction between the H 1s orbital and NM 2pz orbital will split into two orbitals: a fully filled, low-energy bonding orbital (σ) and a partially filled, high-energy antibonding orbital (σ*). As shown in Figure 7a, we can see that with the increasing of P concentrations, the peak changes from sharp to disuse. The peak intensities of TDOS decrease from −4.5 to −3.9 eV and increase from −5.4 to −5.0 eV and −2.6 to −1.0 eV. In Figure 7c, we can see the reason for this change from PDOS of P element 2p orbital and H element 1s orbital. Because of the increasing of P concentrations, the peak intensities of p-DOS increase from −5.7 to −1 eV below Fermi level. From −13.2 to −8 eV, the s-DOS of H element split from one distinct peak into two gentle peaks, which means H element s orbital split. From −4.6 to −5.9 eV the peaks shift to the left and as expected for 0.50 concentration the peak values increase, which indicate the hybridization of s-p orbitals between H-P after absorbing the H atom. Combined with DOS and DCD, we can explain the varied peak values. We find that the position of H and P atoms is changed after structure optimization. When doping 0.50 P element, because of the upward shift of P atoms, more charge transfer occurs between H and P atoms while Ti atoms are less involved. Therefore, the peak value is minimum. When doping 0.75 P element, because H atoms move toward the P rich site, the hybridization of s-p orbitals between H and P increases and gains the maximum peak value. Comparing doping 0.25 and 1.00 P element, the peak value of 1.00 P is larger because more charge is transferred, which is in good agreement with Figure 7c.

From Figure 7b, we can see DOS move to lower energy levels and the peak drop to lower intensities with the increasing of H absorption concentrations. Through PDOS of P element 2p orbital and H element 1s orbital in Figure 7d, we analyzed the reason for this trend. From −13.2 to −8.6 eV and −5.7 to −5.4 eV, the peak values of s-DOS increase, which is due to the increasing of H absorption concentration. Additionally, between −5.7 and −5.4 eV, the peak shift to the left means the energy levels are decreased. The increasing band width shows the decreasing of structure locality. We find that the Fermi level increases with the increasing of H absorption concentration, indicating that the electrical conductivity becomes better. In combination with Figure 8b, we can see that the amounts of charge transfer decreases with the increasing of H absorption concentrations, and the absorption of H atoms on the surface is weakened. Meanwhile, from Figure 8, it is clear that charge transfer does not only occur on the surface, but the subsurface Ti atoms are also involved in it. This explains why the Bader charge is not a good descriptor of charge transfer. The above results suggest that the local structure and global electronic structure regulated by nonmetallic doping synergistically optimizes HER catalytic activity.

### 3.5. Thermodynamic Stability

Understanding the stability of materials is highly necessary for its synthesis and application. Therefore, in order to prove that the candidate catalysts we have screened can be synthesized experimentally, the canonical ensemble (NVT ensemble) ab initio molecular dynamics (AIMD) simulation is performed at 300 K. The temperature fluctuation and structural deformation of Ti_3_C_2_O_2_ and doping 0.25, 0.75 and 1.00 concentrations of P element Ti_3_C_2_O_2_ after running for 3000 fs can be seen in Figure 9. We can see a slight change in the local structure, but the overall morphology remains. Moreover, the temperature fluctuates above and below 300 K shows small fluctuations, and the local structures have no obvious deformation compared to 0 K, which indicates that Ti_3_C_2_O_2_ and doping P element Ti_3_C_2_O_2_ are thermally stable at room temperature and can be synthesized experimentally. 

## 4. Discussion

We investigated the doping nonmetallic element concentrations and H absorption concentration effects on HER activity of Ti_3_C_2_O_2_ by first principles methods. It is found that doping F element can improve the surface stability, while doping P element can improve HER performance. Additionally, by tuning doping concentrations and H absorption concentration, H absorption strength and conductivity can be optimized. When doping 1.00 concentrations P and absorbed 3/4 H, we gained the best HER catalytic activity. Further analysis showed that doping elements regulated the local structure and the overall electronic structure, leading to the interaction of H-NM changing, which is the origin of the improvement of HER catalyst activity. Finally, AIMD analysis proved that P element doped Ti_3_C_2_O_2_ can maintain stability, which is predictive that the catalyst can be synthesized experimentally. This study predicted suitable candidates HER catalysts and provided theoretical guidance for the design of novel HER catalysts.

## Figures and Tables

**Figure 1 nanomaterials-11-02497-f001:**
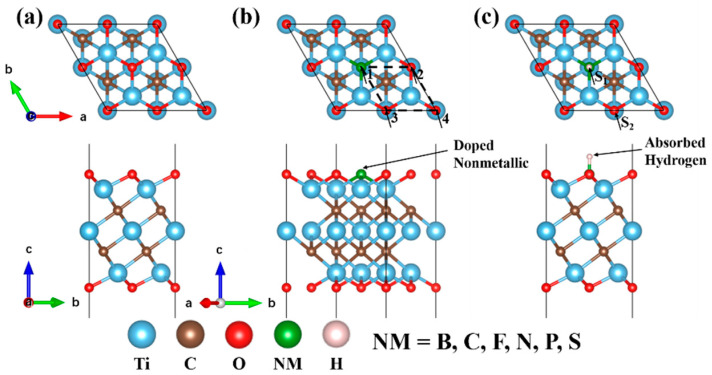
Computational 2 × 2 supercell 2D MXene models. The top one is the top view, and the bottom one is the side view. (**a**) Ti_3_C_2_O_2_ 2 × 2 supercell. (**b**) Nonmetal doped Ti_3_C_2_O_2_ 2×2 supercell; 1, 2, 3 and 4 represent 4 types surface equivalent doping positions. (**c**) H atom adsorbed on nonmetal doped Ti_3_C_2_O_2_ 2 × 2 supercell. S_1_ represent H adsorbed on the doping atom, S_2_ represent H adsorbed on the O atom.

**Figure 2 nanomaterials-11-02497-f002:**
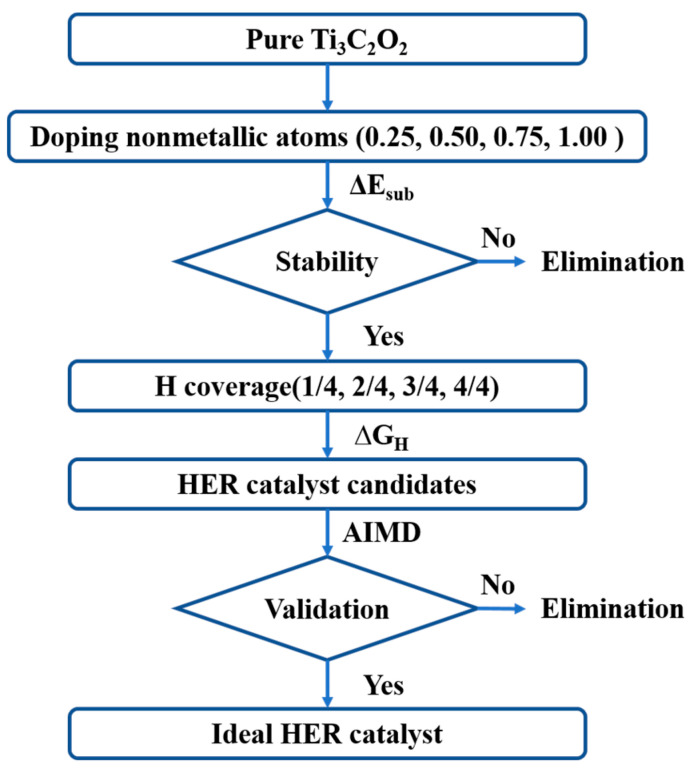
The workflow of calculating and screening nonmetallic element doped Ti_3_C_2_O_2_ HER catalyst.

**Figure 3 nanomaterials-11-02497-f003:**
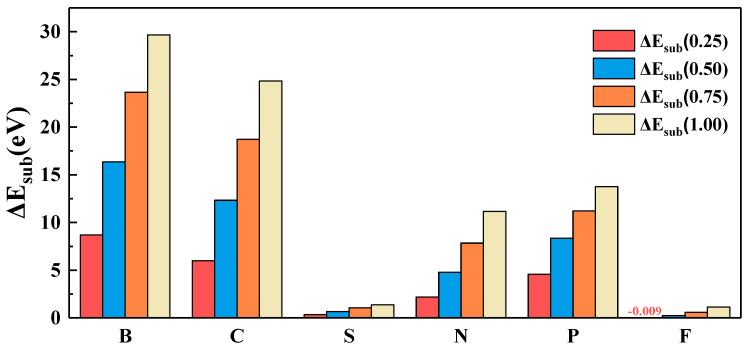
Dopant substitution energy of different nonmetallic elements at different doping concentrations.

**Figure 4 nanomaterials-11-02497-f004:**
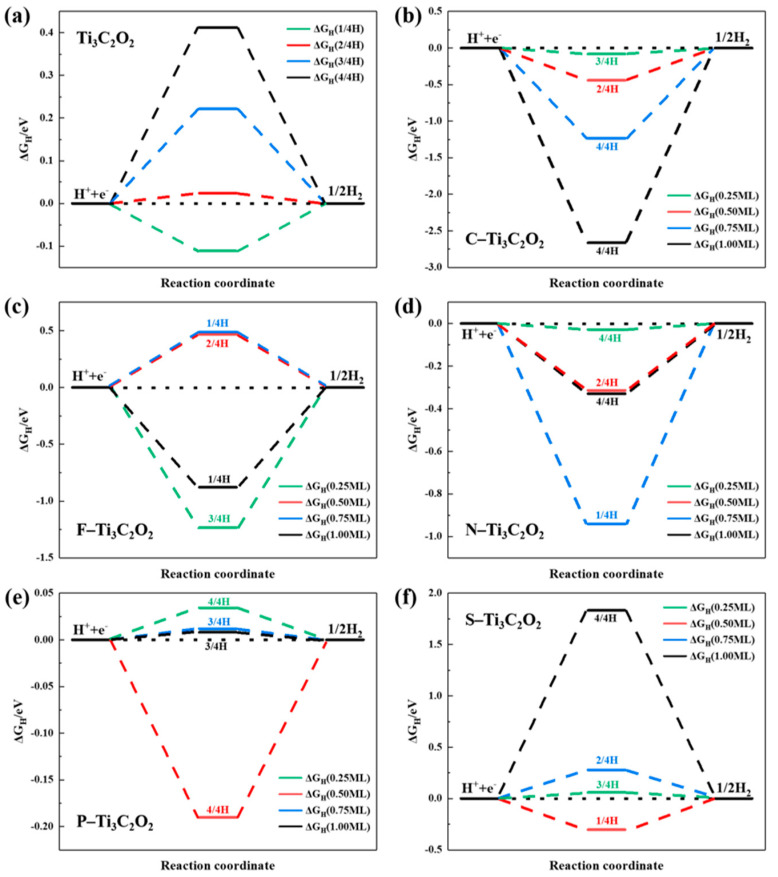
The hydrogen adsorption free energy(∆G_H_) of Ti_3_C_2_O_2_ and nonmetal doping Ti_3_C_2_O_2_. (**a**) Ti_3_C_2_O_2_, (**b**) C doping Ti_3_C_2_O_2_, (**c**) F doping Ti_3_C_2_O_2_, (**d**) N doping Ti_3_C_2_O_2_, (**e**) P doping Ti_3_C_2_O_2_, (**f**) S doping Ti_3_C_2_O_2_.

**Figure 5 nanomaterials-11-02497-f005:**
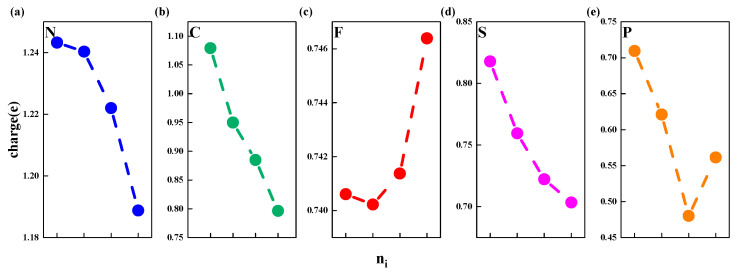
Effect of dopant and surface concentration on the average nonmetal charge; charge means the charge change. (**a**–**e**) represents N, C, F, S, P doped Ti_3_C_2_O_2_, respectively.

**Figure 6 nanomaterials-11-02497-f006:**
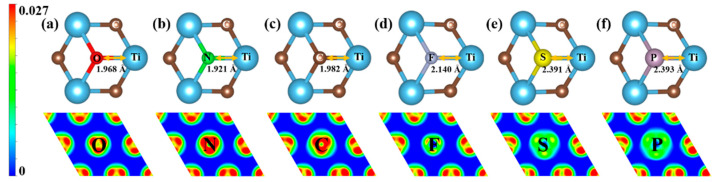
Bond length and the electron density difference. (**a**) Top image shows the local crystal structure and the O-H bond length of 2D Ti_3_C_2_O_2_, the bottom image shows the electron density difference of 2D Ti_3_C_2_O_2_; (**b**–**f**) top image shows the local crystal structure and the NM-H bond length of 2D NM-Ti_3_C_2_O_2_, the bottom image shows the electron density difference of 2D NM-Ti_3_C_2_O_2_. Warm color means gaining electrons, cold color means losing electrons.

**Figure 7 nanomaterials-11-02497-f007:**
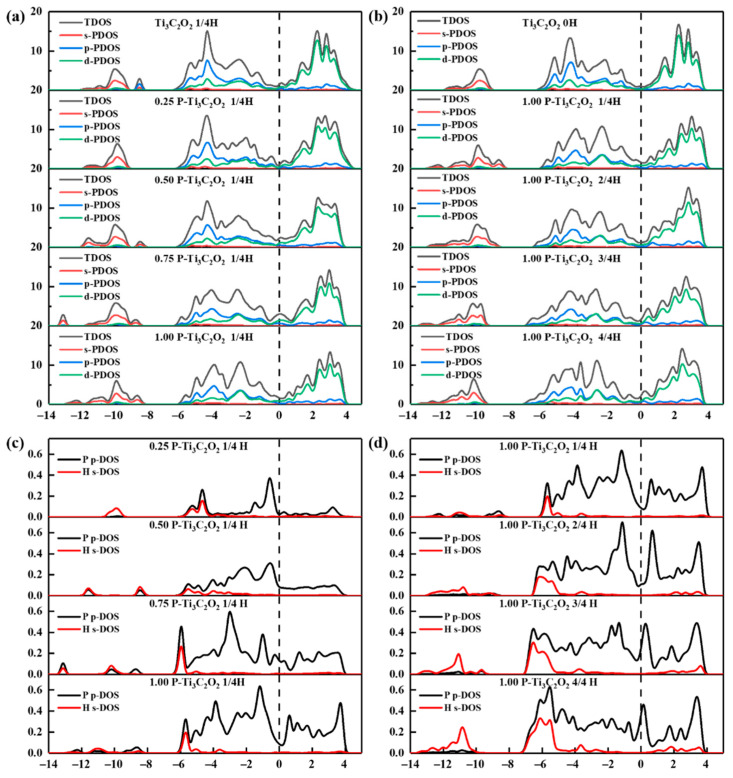
The total density of states (TDOS) and projected density of states (PDOS) of Ti_3_C_2_O_2_ and Ti_3_C_2_O_2_ doped with different concentrations of P elements and absorbed different concentrations H elements. (**a**,**c**) are TDOS and PDOS of Ti_3_C_2_O_2_, 0.25 P-Ti_3_C_2_O_2_, 0.50 P-Ti_3_C_2_O_2_, 0.75 P-Ti_3_C_2_O_2_ and 1.00 P-Ti_3_C_2_O_2_ absorbed 1/4H, respectively; (**b**,**d**) are TDOS and PDOS of Ti_3_C_2_O_2_, and 1.00 P-Ti_3_C_2_O_2_ absorbed 1/4H, 2/4H, 3/4H and 4/4H, respectively.

**Figure 8 nanomaterials-11-02497-f008:**
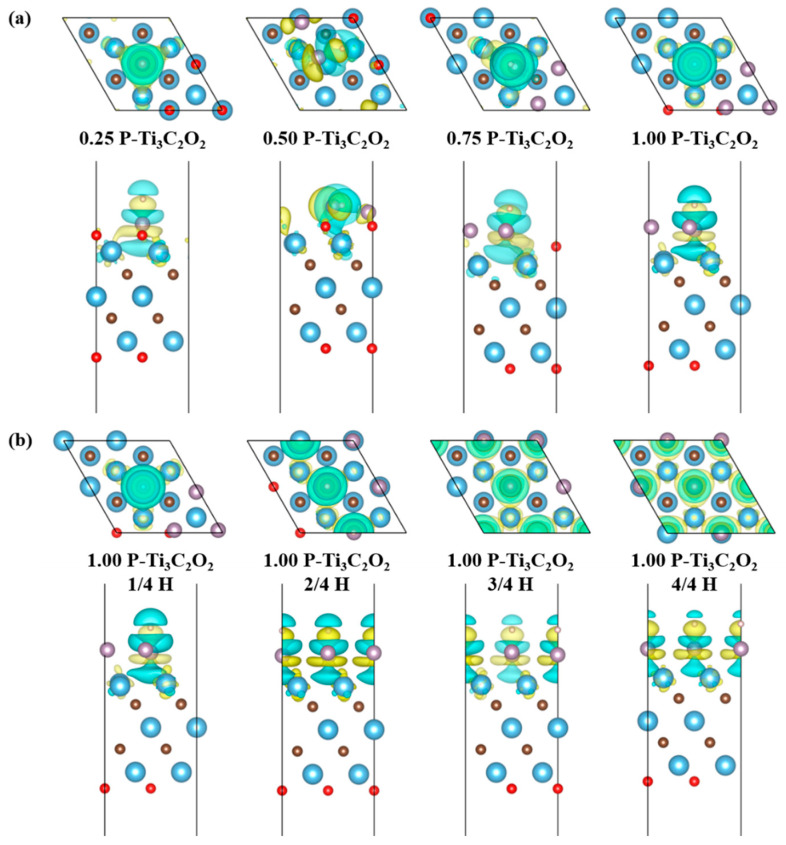
(**a**) The difference charge densities (DCD) of Ti_3_C_2_O_2_ doping 0.25, 0.50, 0.75 and 1.00 concentrations of P element; (**b**) the DCD of Ti_3_C_2_O_2_ doping 1.00 concentrations of P element at 1/4, 2/4, 3/4 and 4/4 H absorption concentrations. Warm color means gaining electrons, cold color means losing electrons.

**Figure 9 nanomaterials-11-02497-f009:**
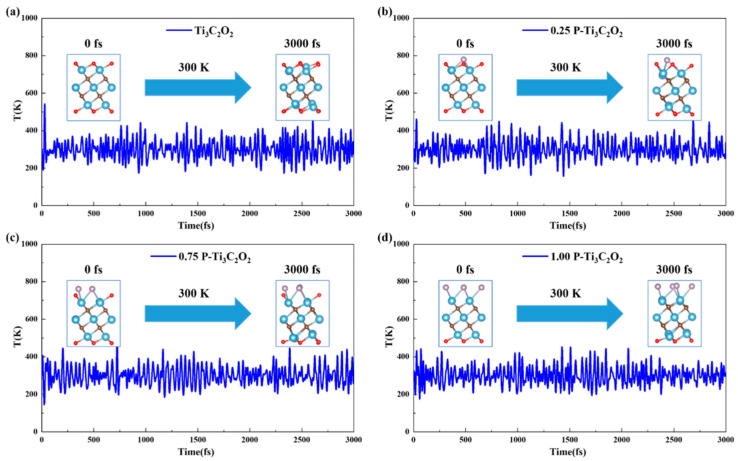
Ab initio molecular dynamics (AIMD) verification: (**a**) AIMD of Ti_3_C_2_O_2_ at 300 K of canonical ensemble (NVT ensemble); (**b**) AIMD of 0.25 P-Ti_3_C_2_O_2_ at 300 K of NVT ensemble; (**c**) AIMD of 0.75 P-Ti_3_C_2_O_2_ at 300 K of NVT ensemble; (**d**) AIMD of 1.00 P-Ti_3_C_2_O_2_ at 300 K of NVT ensemble; the illustrated structure shows the initial and final structural sketch.

## Data Availability

The datasets generated during and/or analyzed during the current study are available from the corresponding author.

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
