# Peer review of "System Theoretical Study on the Effect of Variable Nonmetallic Doping on Improving Catalytic Activity of 2D-Ti3C2O2 for Hydrogen Evolution Reaction"

_nanomaterials, 2021, doi:10.3390/nano11102497_

Round 1

Reviewer 1 Report

In this work, Authors focus on DFT calculations related to nonmetallic doping with B C N F P S atoms to Ti3C2O2 monolayer's surface, their optimal concentration toward stability of the MXenes as well as optimal H concentration for HER catalytic activity and stability.

After being familiarized with others Authors' publications (i.e. J. Mater. Chem. A and Appl. Surf. Sci.) I have no issues with the scientific part of the manuscript. Research were conducted properly, along to Fig.2. Good work.

However, English has to be improved. Used tenses have to be the same according to manuscript parts. There should not be any jumps between tenses like in Introduction. Then, just minor fix of missing comas, missing spaces, double spaces is required.

I recommend this work for publication after text editing.

Author Response

Dear Editors and Reviewers:

Thanks for your valuable comments and suggestions for our work to further improve the manuscript. We have improved the tense and format specification of introduction and manuscript parts. Changes are marked in blue in the revised manuscript.

Thank you very much for all your help and looking forward to hearing from you soon.

Best regards

Sincerely,

Prof. Ping Qian

Beijing Advanced Innovation Center for Materials Genome Engineering

University of Science and Technology Beijing, Beijing 100083, China

Reviewer 2 Report

The manuscript describes the theoretical investigation on the effect of variable nonmetallic doping on improving catalytic activity of 2D-Ti3C2O2 for hydrogen evolution reaction. In my opinion, the current form of the manuscript needs a minor revision. Some of the comments were given below, which need to be addressed before publication.     

  1. The Author stated that (page 7, line 222) “In terms of the type of doping elements, the optimal order of |∆GH| is P (0.008eV) > N (-0.028eV) > S (0.060eV) > C (-0.081eV) > F (0.466eV).” Herein author need to mention the doping percentages for comparison.
  2. The authors stated that (Page 10, line 340) "Moreover, the temperature fluctuates above and below 300K, which means Ti3C2O2 and doping P element Ti3C2O2 can exist stably at room temperature and can be synthesized experimentally. From the figure 9 it is not quite clear what temperature fluctuation author wanted to mean. Author need to justify the statement.
  3. The author estimated the hydrogen adsorption energy (∆GH) of the 2D-Ti3C2O2 upon nonmetal doping, which is general strategy to explain the performance of the catalyst and estimate the active site of catalyst. However, the performance and durability of the catalyst depends on the pH of the medium. What is author opinion on pH effect on ∆GH.
